# LEADER: Learning Attention over Driving Behaviors for Planning under Uncertainty

**Mohamad H. Danesh** [*]
Oregon State University
daneshm@oregonstate.edu

**Panpan Cai** [†]
Shanghai Jiao Tong University
cai_panpan@sjtu.edu.cn

**David Hsu**
National University of Singapore
dyhsu@comp.nus.edu.sg

**Abstract:** Uncertainty in human behaviors poses a significant challenge to autonomous driving in crowded urban environments. The *partially observable Markov decision process* (POMDP) offers a principled general framework for decision making under uncertainty and achieves real-time performance for complex tasks by leveraging Monte Carlo sampling. However, sampling may miss rare, but critical events, leading to potential safety concerns. To tackle this challenge, we propose a new algorithm, *LEarning Attention over Driving bEhavioRs* (LEADER), which learns to attend to critical human behaviors during planning. LEADER learns a neural network generator to provide attention over human behaviors; it integrates the attention into a belief-space planner through importance sampling, which biases planning towards critical events. To train the attention generator, we form a minimax game between the generator and the planner. By solving this minimax game, LEADER learns to perform risk-aware planning without explicit human effort on data labeling. [3]

## 1 Introduction

Robots operating in public spaces often contend with a challenging crowded environment. A representative is autonomous driving in busy urban traffic, where a robot vehicle must interact with many human traffic participants. A significant challenge is posed by the vast amount of *uncertainty* in human behaviors, e.g., on their intentions, driving styles, etc.. The *partially observable Markov decision processes* (POMDPs) [1] offer a principled framework for planning under uncertainty. However, optimal POMDP planning is computationally expensive. To achieve real-time performance for complex problems, practical POMDP planners [2, 3] often leverage *Monte-Carlo (MC) sampling* to make approximate decisions, i.e., sampling a subset of representative future scenarios, then condition decision-making on the sampled future. They have shown success in various robotics applications, including autonomous driving [4], navigation [5, 6], and manipulation [7, 8].

Sampling-based POMDP planning, however, may compromise safety. It is difficult to sample future events with low probabilities, and some may lead to disastrous outcomes. In autonomous driving, rare, but critical events often arise from adversarial human behaviors, such as recklessly overtaking the ego-vehicle or making illegal U-turns. Failing to consider them would lead to hazards. Earlier work Luo et al. [9] indicates that re-weighting future events using *importance sampling* leads to improved performance. The approach requires an *importance distribution* (IS) for sampling events. In autonomous driving, an importance distribution re-weights the different *intentions* of human participants, e.g., routes they intend to take. A higher weight means an increased probability of sampling an intended route, thus examining its consequence more carefully during planning. Prior works have used IS to better verify the safety of driving policies and proposed ways to synthesize

---

[*]Work done when interning at National University of Singapore

[†]Corresponding author: Panpan Cai (cai_panpan@sjtu.edu.cn)

[3]Code is available at: https://github.com/modanesh/LEADER

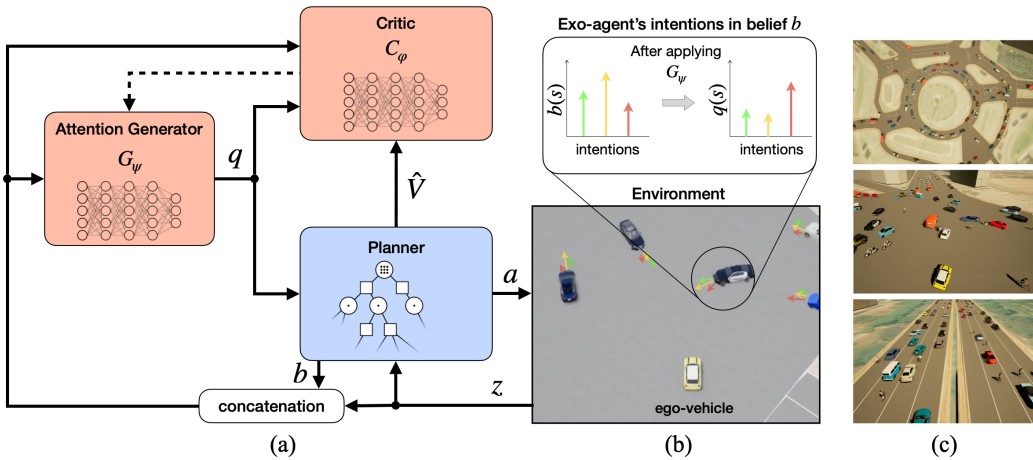

Figure 1: Overview of LEADER. (a) LEADER contains a learning component (red) and a planning component (blue). The learning components include: an attention generator $G_\psi$ that tries to generate attention $q$ over human behaviors, based on the current belief $b$ and observation $z$ from the environment; and a critic $C_\varphi$ that approximates the planner's value estimate, $\hat{V}$, based on $b$, $z$ and the generated attention, $q$. The planning component performs risk-aware planning using the learned attention $q$. It decides an action $a$ to be executed in the environment and collects experience data. (b) Attention is defined as an importance distribution over human behavioral intentions. The upper box shows the probability of different intentions of the highlighted exo-agent in green, yellow and red, as well as how the attention generator maps the natural-occurrence probabilities to importance probabilities, by highlighting the most adversarial intention (red). (c) We train LEADER using three simulated real-life urban environments: Meskel Square in Addis Ababa, Ethiopia, Magic Roundabout in Swindon, UK, and Highway in Singapore.

importance distributions offline [10, 11, 12, 13]. However, it is difficult to obtain importance distributions for online planning, due to the real-time constraints.

We propose a new algorithm, LEADER, which learns importance distributions both *from* and *for* sampling-based POMDP planning. The algorithm uses a neural network *attention generator* to obtain importance distributions over human behavioral intentions for real-time situations. Despite the neural networks' problem in terms of interpretability [14], they are quite useful in problems that need to be scaled, such as crowd driving. Next, an online POMDP planner consumes the importance distribution to make risk-aware decisions, applying importance sampling to bias reasoning towards critical human behaviors. To train the algorithm, we treat the attention generator and the planner as opponents in a minimax game. The attention generator seeks to *minimize* the planner's *value*, or the expected cumulative return. This maps to highlight the most adversarial human behaviors by learning from experience. On the other hand, the planner seeks to *maximize* the value conditioned on the learned attention, which maps to find the best conditional policy using look-ahead search. By solving the minimax game, the algorithm learns to perform risk-aware planning, without human labeling. In our experiments, we evaluate the performance of LEADER for autonomous driving in dense urban traffic. Results show that LEADER significantly improves driving performance in terms of safety, efficiency and smoothness, compared to both risk-aware planning and risk-aware reinforcement learning.

## 2 Background and Related Work

### 2.1 POMDP Planning and Monte Carlo Sampling

The Partially Observable Markov Decision Process (POMDP) [1] models the interaction between a robot and an environment as a discrete-time stochastic process. A POMDP is written as a tuple $\langle S, A, Z, T, R, O, \gamma, b_0 \rangle$ with $S$ representing the space of all possible states of the world, $A$ denoting the space of all possible actions the robot can take, and $Z$ as the space of observations it can receive. Function $T(s, a, s') = p(s'|s, a)$ represents the probability of the state transition from $s \in S$ to

$s' \in S$ by taking action $a \in A$. The function $R(s, a)$ defines a real-valued reward specifying the desirability of taking action $a \in A$ at state $s \in S$. The observation function $O(s', a, z) = p(z|s', a)$ specifies the probability of observing $z \in Z$ by taking action $a \in A$ to reach to $s' \in S$. $\gamma \in [0, 1)$ is the discount factor, i.e., the rate of reward deprecation over time. Because of the robot's perception limitations, the world's full state is unknown to the robot, but can be inferred in the form of *beliefs*, or probability distributions over $S$. The robot starts with an initial belief $b_0$ at $t = 0$, and updates it throughout an interaction trajectory using the Bayes rule [15], according to the actions taken and observations received. POMDP planning searches for a closed-loop policy, $\pi^* : B \rightarrow A$, prescribing an action for any belief in the belief space $B$, which maximize the policy *value*, $V_\pi(b_0) = \mathbb{E}\left[\sum_{t=0}^{\infty} \gamma^t R(s_t, \pi(b_t))|b_0, \pi\right]$, which computes the cumulative reward to be achieved in the current and future time steps, $t \geq 0$, if the robot chooses actions according to policy $\pi$ and the updated belief $b_t$, from the initial belief $b_0$ onwards. Discounting using $\gamma \in [0, 1)$ keeps the value bounded.

Online POMDP planning often performs look-ahead search, constructing a belief tree starting from the current belief and branching with future actions and observations. Enumerating all possible futures, however, is often computationally intractable. Practical algorithms like DESPOT [2] leverage Monte-Carlo sampling and heuristic search to break the computational difficulty. DESPOT samples initial states and future trajectories using the POMDP simulative model. Denote a trajectory as $\zeta = (s_0, a_1, s_1, z_1, a_2, s_2, z_2, ...)$. The initial state $s_0$ is sampled from the current belief $b$. Given any subsequent state $s_t$ and robot action $a_t$, the next state $s_{t+1}$ and the observation $z_{t+1}$ are sampled with a probability of $p(s_{t+1}, z_{t+1}|s_t, a_{t+1}) = O(s_{t+1}, a_t, z_{t+1})T(s_t, a_t, s_{t+1})$. A DESPOT tree collates a set of sampled trajectories to approximately represent the future. Each node in the tree contains a set of sampled future states, forming an approximate future belief. The DESPOT tree branches with all possible actions and then sampled observations under each visited belief, effectively considering all candidate policies under the sampled scenarios. Figure 2(a) shows an example DESPOT tree.

DESPOT evaluates the value of a policy using Monte-Carlo estimation:

$$V_\pi(b) \quad = \quad \int_{\zeta \sim p(\cdot|b,\pi)} V_\zeta d\zeta \approx \sum_{\zeta \in \Delta} p(\zeta|b, \pi)V_\zeta \tag{1}$$

where $\Delta$ is a set of trajectories sampled by applying $\pi$. Also,

$$p(\zeta|b, \pi) = b(s_0) \prod_{t=0}^{H-1} p(s_{t+1}, z_{t+1}|s_t, a_{t+1}) \tag{2}$$

is the probability of a trajectory $\zeta$ being sampled, $V_\zeta = \sum_{t=0}^{H-1} \gamma^t R(s_t, a_{t+1})$ is the cumulative reward along $\zeta$, and $H$ is maximum look-ahead depth, or the *planning horizon*. By incrementally building a belief tree using sampled trajectories, DESPOT searches for the policy that provides the best value estimate, and outputs the optimal action for $b$ when exhausting the given planning time.

### 2.2 Risk-Aware Planning and Learning Approaches

There are different techniques for risk-aware planning under uncertainty. Nyberg et al. [16] and Gilhuly et al. [17] proposed two measures for estimating safety risks along driving trajectories, with a focus on open-loop trajectory optimization. Huang et al. [18] modeled risk-aware planning as a chance-constrained POMDP to compute closed-loop policies that provide a low chance of violating safety constraints. Kim et al. [19] leveraged bi-directional belief-space solvers. It bridges forward belief tree search with heuristics produced by an offline backward solver. Both methods improved the performance of POMDP planning in safety-critical domains, but at the cost of an expensive offline planning stage. Thus, it can hardly apply to large-scale problems. Most relevant to our method, Luo et al. [9] offered IS-DESPOT, which improves the performance of online POMDP planning by leveraging importance sampling (IS). It biases MC sampling towards critical scenarios according to an importance distribution provided by human experts, then computes a risk-aware policy under the re-weighted scenarios. Manually constructing the importance distribution, however, is difficult for complex problems such as driving in an urban crowd. In this paper, we propose a principled approach to learn importance distributions from experience and adapt them with real-time situations.

LEADER is also loosely connected to risk-aware reinforcement learning (RL). Kamran et al. [20] proposed a risk-aware Q-learning algorithm by punishing risky situations instead of only collision failures. Mirchevska et al. [21] proposed to combine DQN with a rule-based safety checker, masking

out infeasible actions. Eysenbach et al. [22] offered Ensemble-SAC (ESAC) for risk-aware RL, by additionally learning an ensemble of reset policies to assist the robot avoid irreversible states during training. However, because of low sample efficiency, the above model-free approaches were limited to small-scale tasks like lane changing in regulated highway traffic or controlling articulated robots for simple simulated tasks. Some prior work improved the risk-awareness of model-based RL that plans with learned models. Thomas et al. [23] proposed SMBPO, which learns a sufficiently-large terminal cost for failure states. Berkenkamp et al. [24] focused on ensuring the stability of control. The model-based methods provided better sample efficiency. However, it is still difficult to learn a model for large-scale, safety-critical problems like driving in an urban crowd [25].

### 2.3   Integrating Planning and Learning

LEADER integrates planning and learning to benefit from both explicit reasoning and past experience. It uses learning to assist explicit POMDP planning, which has been inspired by the following prior works. The LeTS-Drive algorithm [26, 27] first proposed to learn heuristics for solving POMDPs, with planner actors and the learner collaborating in an closed-loop. Later, Lee et al. [28] proposed a generator-critic framework to learn macro-actions for POMDPs. It uses a neural network critic to approximate the planner's value function and enable end-to-end training. This work extends the generator-critic framework, applying it to solve a min-max game for learning behavioral attentions.

## 3   Overview

LEADER learns to attend to the most critical human behaviors for risk-aware planning in urban driving. We define attention over the behavioral intentions of human participants. Assume there are $N$ *exo-agents* (traffic participants) near the robot. Each of them may undertake a finite set of *intentions*, or future routes such as keeping straight, turning left, merging into the right, etc, extracted from the road context by searching the lane network. The actual intention of an exo-agent is not directly observable. At each time step, LEADER maintains a belief $b$ and an importance distribution $q$ over the intention sets of the $N$ exo-agents. The belief $b$ specifies the *natural occurrence probability* of exo-agents' intentions. It is inferred from the interaction history. The importance distribution $q$ specifies the *attention* over exo-agents' intentions, determining the actual probability of sampling them during planning. We propose to learn the importance distribution or attention mechanism from the experience of an online POMDP planner. We will use "importance distribution" and "attention" interchangeably in the remaining.

LEADER has two main components: a learner generating attention for real-time situations, and a planner consuming attention to perform conditional planning. The generator uses a neural network, $q = G_\psi(b, z)$, to generate an importance distribution $q$, for the real-time situation specified by the current belief $b$ and observation $z$. The planner uses online belief tree search to make risk-aware decisions for given $b$ and $z$, leveraging the importance distribution $q$ to bias Monte-Carlo sampling towards critical human behaviors. To train LEADER, generator and planner form a minimax game:

$$\min_{q \in Q} \max_{\pi \in \Pi} \hat{V}_\pi(b, z|q) \tag{3}$$

where $\Pi$ is the space of all policies, and $Q$ is the space of all importance distributions. In this game, the generator must learn to generate $q$'s that lead to the lowest planning value, meaning to increase the probability of sampling the most adversarial intentions of exo-agents; the planner must find the best policy with the highest value, conditioned on the generated $q$. We further learn a critic function, $v = C_\varphi(b, z, q)$, another neural network that approximates the value function of the planner, $C_\varphi(b, z|q) \approx \max_{\pi \in \Pi} \hat{V}_\pi(b, z|q)$, to assist gradient-descent training of the generator. Figure 1a and 1b demonstrate the training architecture of LEADER. In the bottom row, the planner plans robot actions using the generated attention and feeds the driving experience to a replay buffer. In the top row, we train the critic and the generator using sampled data from the replay buffer. The critic is fitted to the planner's value estimates using supervised learning; the generator is trained to maximize the planner's value, using the critic as a differentiable surrogate objective.

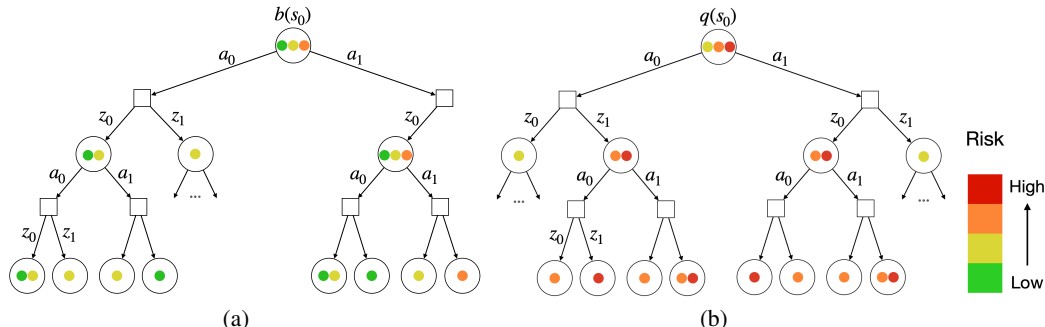

Figure 2: A comparison of DESPOT and the LEADER planner. (a) The DESPOT tree samples intentions from the belief, $b(s_0)$, without considering the criticality of the intentions. Some rare critical events might be missed during sampling. (b) The LEADER tree samples intentions from the learned importance distribution, $q(s_0)$, which is biased towards adversarial intentions. The tree thus considers more critical events (red), and less safe events (green).

## 4    Risk-Aware Planning using Learned Attention

The LEADER planner performs belief tree search to plan robot actions conditioned on the attention over exo-agents' behaviors, solving the inner maximization problem in Eq. (3). A belief tree built by LEADER looks similar to a DESPOT tree (Section 2.1). It collates many sampled trajectories, each corresponding to a top-down path in the tree. The tree branches over all actions and all sampled observations under each belief node, effectively considers all possible policies under the sampled scenarios. The difference is, however, LEADER biases the tree towards higher-risk trajectories using importance sampling. Figure 2 provides a side-by-side comparison of belief trees in DESPOT and LEADER. Appendix A introduces the basics of importance sampling, and a theoretical justification on our minimization objective of learning importance distributions.

Concretely, we sample initial states $s_0$ from the learned importance distribution $q(s_0)$, instead of from the actual belief $b(s_0)$. As a result, the sampling distribution of simulation trajectories is also altered from Eq. (2), becoming:

$$q(\zeta|b, z, \pi) = q(s_0) \prod_{t=0}^{D-1} p(s_{t+1}, z_{t+1}|s_t, a_{t+1}), \tag{4}$$

where $\zeta$ is a hypothetical future trajectory. The value of a candidate policy $\pi$ is now evaluated as:

$$V_\pi(b) \quad = \quad \int_{\zeta \sim p(\cdot|b,z,\pi)} V_\zeta d\zeta = \int_{\zeta \sim q(\cdot|b,z,\pi)} \frac{p(\zeta|b, z, \pi)}{q(\zeta|b, z, \pi)} V_\zeta d\zeta \tag{5}$$

Here, $V_\zeta$ is the discounted cumulative reward along a trajectory $\zeta$. Eq. (5) first shows the definition of the value of policy $\pi$, then applies importance sampling, replacing the sampling distribution $p(\cdot|b, z, \pi)$ in Eq. (2) with $q(\cdot|b, z, \pi)$ in Eq. (4). It also uses the importance weights $\frac{p(\zeta|b,z,\pi)}{q(\zeta|b,z,\pi)}$ to unbias the value estimation and ensure the correctness of planning. The value is further approximated using Monte Carlo estimates:

$$\hat{V}_\pi(b) \quad = \quad \frac{1}{|\Delta'|} \sum_{\zeta \in \Delta'} \frac{p(\zeta|b, z, \pi)}{q(\zeta|b, z, \pi)} V_\zeta = \frac{1}{|\Delta'|} \sum_{\zeta \in \Delta'} \frac{p(s_0)}{q(s_0)} V_\zeta, \tag{6}$$

Eq. (6) starts with approximating the value using a set of sampled trajectories, $\Delta'$. It then simplifies the importance weights to $\frac{p(s_0)}{q(s_0)}$, as $p(\zeta|b, z, \pi)$ and $q(\zeta|b, z, \pi)$ only differ in the probability of sampling the initial state $s_0$.

The LEADER planner is built on top of IS-DESPOT [9], integrating it with learned importance distributions. Following IS-DESPOT, LEADER performs anytime heuristics search, incrementally building a sparse belief tree when sampling more trajectories. During the search, it maintains for each belief node a set of approximate value estimates, and uses them as tree search heuristics. See Luo et al. [9] for more details of the anytime algorithm.

# 5   Learning Attention over Human Behaviors

We train the critic and generator neural networks using driving experience from the planner, stored in the replay buffer. The generator is trained to minimize the planner's value estimates, solving the outer minimization problem in Eq. (3). It uses the critic as a differentiable surrogate objective, which is supervised by the planner's value estimates. Appendix B describes the network architectures of our critic and generator.

**Critic Network**. The Critic network's parameters $\varphi$ are updated by minimizing the L2-norm between the critic's value estimate and the planner's value estimate $\hat{V}$ using gradient descent, given a sampled tuple of belief $b$, observation $z$, and attention $q$:

$$J(\varphi) = \mathbb{E}_{(b,z,q,\hat{V})\sim D}[|C_\varphi(b,z,q) - \hat{V}(b,z|q)|^2], \tag{7}$$

where $D$ is the set of online data stored in the replay buffer.

**Attention Generator Network**. The generator network's parameters $\psi$ are updated by minimizing the planner's value as estimated by the critic, given a sampled tuple of belief $b$ and observation $z$:

$$J(\psi) = \mathbb{E}_{(b,z)\sim D}\Big[\mathbb{E}_{q\sim G_\psi(b,z)}[C_\varphi(b,z,q)]\Big] \tag{8}$$

where $D$ represents online data stored in the replay buffer. This objective is made differentiable using the reparameterization trick [29], enabling gradient descent training via the chain rule:

$$J(\psi) = \mathbb{E}_{(b,z)\sim D}\Big[\mathbb{E}_{\epsilon\sim\mathcal{N}(0,1)}[C_\varphi\big(b,z,G_\psi(b,z,\epsilon)\big)]\Big] \tag{9}$$

The following is the training procedure of LEADER. In each time step, the current belief $b$ and observation $z$ are fed into $G_\psi$ to produce the importance distribution $q$. Then, the planner takes $b$, $z$ and $q$ as inputs to perform risk-aware planning, and outputs the optimal action $a$ to be executed in the environment together with its value estimate $\hat{V}$. The data point $(b,z,q,\hat{V})$ is sent to a fixed-capacity replay buffer. Next, a batch of data is sampled from the replay buffer, and used to update $C_\varphi$ and $G_\psi$ according to Eq. (7) and (8). The updated $G_\psi$ is then used for next planning step. Training starts from randomly initialized generator and critic networks and an empty replay buffer. In the warm-up phase, the critic is first trained using data collected by LEADER with uniform attention. This provides meaningful objectives for the attention generator to start with. Then, both the critic and the generator are trained with online data collected using the latest attention generator. At execution time, we only deploy the generator and the planner to perform risk-aware planning.

# 6   Experiments and Discussions

We evaluate LEADER on autonomous driving in unregulated urban crowds, show the improvements on the real-time driving performance, and analyze the learned attention. The experiment task is to control the acceleration of a robot ego-vehicle, so that it follows a reference path and drives as fast as possible, while avoiding collision with the traffic crowd under the uncertainty of human intentions. A human intention indicates which path one intends to take. Candidate paths are extracted from the lane network of the map, according to the current location of the exo-agent. Attention is thus defined as a joint importance distribution over the intentions of 20 exo-agents near the robot, assuming conditional independence between agents. See details on the POMDP model in Appendix C.

Our baselines include both risk-aware planning and risk-aware RL methods. We first compare with POMDP planners using handcrafted attention. DESPOT-Uniform uses DESPOT with uniform attention over human intentions [30]. DESPOT-TTC computes the criticality of exo-agent's intentions using the estimated Time-To-Collision (TTC) with the ego-vehicle. The attention score is set proportional to $\frac{1}{t_c}$, where $t_c$ is the TTC if the exo-agent takes the indented path with a constant speed. For RL baselines, we first include a model-free RL method, Ensemble SAC (ESAC) [22], which learns an ensemble of Q-values for risk-aware training. Then, we include a model-based RL method, SMBPO [23], which plans with learned dynamics and a risk-aware reward model. Because of the difficulty of real-world data collection and testing for driving in a crowd, we instead used the SUMMIT simulator [30], which simulates high-fidelity massive urban traffic on real-world maps, using a realistic traffic motion model [31]. We evaluate LEADER and all baselines on three different

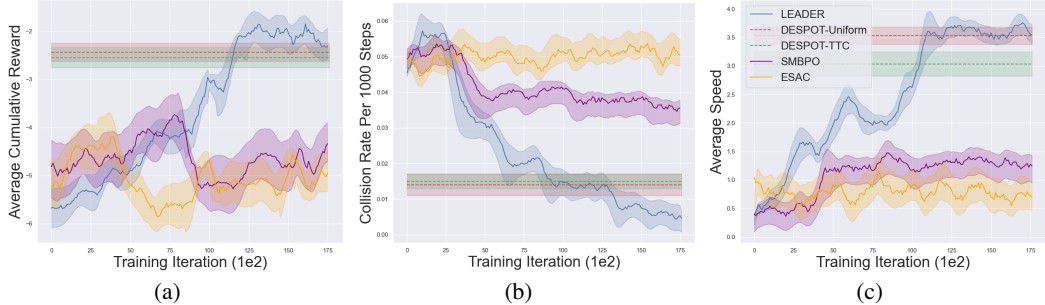

Figure 3: Learning curves of LEADER and existing risk-aware baselines: (a) average cumulative reward (b) collision rate per 1000 steps (c) average speed.

Table 1: Generalized driving performance of LEADER and existing risk-aware planning and learning algorithms on 400 test runs in novel scenes. Errors represent two standard deviations.

| Algorithm | Cumulative Reward | Collision Rate | Travelled Distance | Smoothness Factor |
|---|---|---|---|---|
| DESPOT-Uniform | $-2.44 \pm 0.11$ | $0.014 \pm 0.003$ | $102.12 \pm 5.2$ | $3.31 \pm 0.03$ |
| DESPOT-TTC | $-2.61 \pm 0.14$ | $0.016 \pm 0.002$ | $100.11 \pm 6.34$ | $3.15 \pm 0.12$ |
| ESAC | $-5.8 \pm 0.43$ | $0.05 \pm 0.002$ | $19.38 \pm 5.94$ | $1.72 \pm 0.23$ |
| SMBPO | $-4.91 \pm 0.47$ | $0.038 \pm 0.002$ | $47.2 \pm 12.49$ | $2.16 \pm 0.2$ |
| LEADER (ours) | $\mathbf{-2.1 \pm 0.16}$ | $\mathbf{0.007 \pm 0.001}$ | $\mathbf{115.29 \pm 6.1}$ | $\mathbf{4.64 \pm 0.07}$ |

environments: the Meskel square in Addis Ababa, a highway in Singapore, and the magic roundabout in Swindon, using 5 different sets of reference paths and crowd initialization for each map. Crowd interactions during driving are perturbed with random noise, thus are unique for each episode. See sample driving videos of LEADER here: https://sites.google.com/view/leader-paper.

### 6.1 Performance Comparison

**Learning Curves.** Figure 3 shows the learning curves of LEADER and the baseline algorithms in terms of the average cumulative reward, the collision rate, and the average driving speed. LEADER starts to outperform the strongest RL baseline from the $8500th$ iteration which is 12 hours of training using 12 real-time planner actors. It starts to outperform the strongest planning baseline from the $12500th$ iteration corresponding to 19 hours of training. At the end of training, LEADER achieves the highest cumulative reward, the lowest collision rate, and the highest driving speed among algorithms.

**Generalization.** To evaluate the generalization of LEADER, we ran it with 5 unseen reference paths and crowd initialization for each map with randomness on crowd interactions. Table 1 shows detailed driving performance of the considered algorithms averaged over 400 test runs, equally apportioned among the three maps and map setups. Evaluation criteria include the cumulative reward, collision rate, travelled distance, and smoothness factor. The collision rate measures the average number of collisions per 1000 time steps. The smoothness factor is the reverse of the number of decelerations, $\frac{1}{N_{dec}}$. As shown, LEADER outperforms other methods in all metrics, consistent with the learning curves. It drives much more safely and efficiently than RL methods which had a hard time handling the highly dynamic crowd. It also improves driving safety, efficiency and smoothness from the planning baselines which applied sub-optimal attention.

### 6.2 The Learned Attention

To provide a qualitative analysis on what LEADER has learned, we further provide 2D visualizations of the learned attention in Figure 4. We provide three scenes from the Singapore Highway, Meskel Square, and Magic Roundabout, respectively. In each scene, we highlight a representative exo-agent, show the set of intention paths and visualize the learned attention over the paths using color coding. For other exo-agents, we only show the most critical intentions with the highest attention scores.

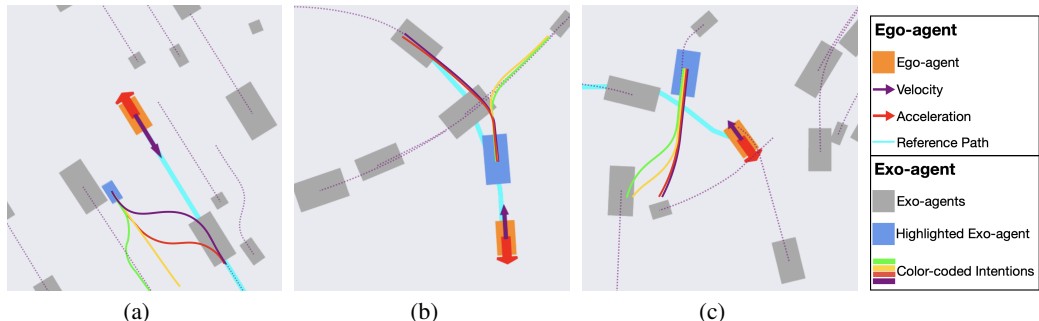

Figure 4: Visualization of the learned attention in: (a) highway (b) Meskel square (c) Magic. We highlighted one exo-agent in blue in each scene. Learned attention over its intentions are color-coded: green, yellow, red, purple, sorted from low attention to high attention. For other exo-agents, we only show the most-attended intention with dotted lines. See more examples here: https://sites.google.com/view/leader-paper.

In Figure 4 (a), the highlighted exo-agent in blue drives in parallel with the ego-vehicle. It has several possible intentions, including continuing straight and merging to neighbouring lanes. The intention of merging right is more likely, as the agent is closer to the right lane. However, the merging left intention is more critical, as the path will interfere with the ego-vehicle's. LEADER learns to put more attention on the more critical possibility. As a result, the planner decides to slow down the vehicle and prepare for potential hazards. In Figure 4 (b), the blue agent can either proceed left or turn right. Both intentions are equality likely. While the turning right intention gets the agent out of the ego-vehicle's way, proceeding left, however, causes the agent to continue interacting with the ego-vehicle. The generator thus assigned more attention to the latter. The planner subsequently decides to stop the ego-vehicle, since the blue agent in front may need to wait for its own path to be cleared. In Figure 4 (c), all intention paths of the blue agent intersect with the ego-vehicle's. The generator learned to attend to the path closest to the ego-vehicle, which is more hazardous. Consequently, the planner stops the ego-vehicle to prevent collision.

## 7 Summary, Limitations and Future Work

In this paper, we introduced LEADER, which integrates learning and planning to drive in crowded environments. LEADER forms a minimax game between a learning component that generates attention over potential human behaviors, and a planning component that computes risk-aware policies conditioned on the learned attention. By solving the minimax game, LEADER learned to attend to the most adversarial behaviors and perform risk-aware planning. Our results show that LEADER helps to improve the real-time performance of driving in crowded urban traffic, effectively lowering the collision rate while keeping the efficiency and smoothness of driving, compared to risk-aware planning and learning approaches.

Limitations of this work are related to assumptions, model errors and data required for training. This work makes a few assumptions. First, we assume there is an available map of the urban environment, providing lane-level information. However, we believe this requirement can be met for most major cities. Second, in our POMDP model, we focused on modeling the uncertainty of human intentions, and ignored perception uncertainty such as significant observation noises and occluded participants. These aspects will be addressed in future work through building more comprehensive models. Third, LEADER will also be affected by model errors, as the approach relies on the planner's value estimates to provide learning signals. But since our approach emphasizes the most adversarial future, i.e., relies on conservative predictions, it is more robust to model errors than typical planning approaches. Lastly, although we have improved tremendously from the sample-efficiency of RL algorithms, LEADER still requires hours of online training. A promising solution is to "warm up" the critic and generator networks using offline real-world driving datasets, such as the Argoverse Dataset [32] and the Waymo Open Dataset [33], then perform further online training.

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

# Appendix

## A   Theoretical Background of Importance Sampling

Importance sampling is a variance-reduction technique for Monte Carlo sampling, often used to obtain more reliable estimations from fewer samples. Given a random variable $x \sim p(x)$ and a function $f(x)$. Suppose we want to estimate the expected value of $f(x)$ with sampling:

$$\mu = \mathbb{E}[f(x)] = \int f(x)p(x)dx \approx \frac{1}{n}\sum_i^n f(x_i), \tag{10}$$

where $\{x_i\}_{i=1,\dots,n}$ are sampled from $p(x)$, and $\frac{1}{n}\sum_i^n f(x_i)$ is the estimation using the $n$ samples. Given an *importance distribution* $q(x)$, importance sampling is performed as:

$$\mu = \int f(x)p(x)\frac{q(x)}{q(x)}dx \approx \frac{1}{n}\sum_i^n f(x_i)\frac{p(x_i)}{q(x_i)}, \tag{11}$$

where $\{x_i\}_{i=1,\dots,n}$ are now sampled from the importance distribution $q(x)$, and $\frac{p(x_i)}{q(x_i)}$, often referred to as the *importance weights*, are used to correct the point-based estimation. If we define $\hat{\mu}_q = \sum_i^n f(x_i)\frac{p(x_i)}{q(x_i)}$, $\hat{\mu}_q$ is an unbiased estimator of $\mu$.

It has also been shown that the variance of $\hat{\mu}_q$ is,

$$Var[\hat{\mu}_q] = \frac{1}{n}\int \frac{(f(x)p(x) - \mu q(x))^2}{q(x)}dx. \tag{12}$$

Therefore, the optimal importance distribution $q^*$ that offers the lowest variance is:

$$q^*(x) = \frac{|f(x)|p(x)}{\mu} \tag{13}$$

The above equation suggests increasing the sampling probability of those $x$ with high absolute function values $|f(x)|$. This theoretical indication supports our core idea of learning to emphasize *critical* events that lead to hazards and thus large negative values.

## B   Neural Network Details

Figure 5 demonstrates network architectures of the attention generator and the critic. The attention generator network first concatenates numbers in the current belief $b$ and the observation $z$ into a single vector and feeds it to a feature extractor. The feature extractor consists of 10 fully-connected layers with ReLU activation. The extracted features are input to a Gated Recurrent Unit (GRU) cell [34], which keeps track of history inputs in its latent memory. Based on the latent memory, the network uses two fully-connected layers and a soft-max layer to output the importance distribution $q$. The critic network has similar architecture. The belief $b$, observation $z$, and the generated importance distribution $q$ are first concatenated into a single vector. Then, the vector is fed to a feature extractor containing 6 fully-connected layers with ReLU activation, then input to a GRU cell for tracking memory. Based on the latent memory, the critic network uses another 3 fully-connected layers to finally output a single decimal number, representing the estimated planner value $v$. See detailed representations of $b$, $z$, and $q$ in the following section.

## C   The POMDP Model for Urban Driving

Our POMDP model for driving in an ill-regulated dense urban traffic is defined as follows:

- **State Modeling**: A world state $s$ encodes:
  - the state of the ego-vehicle, $s_c = (p_c, v_c, \alpha_c, P_c)$, where $p_c$, $v_c$ and $\alpha_c$ denote its position, velocity, and heading direction, and $P_c$ denotes its reference path.

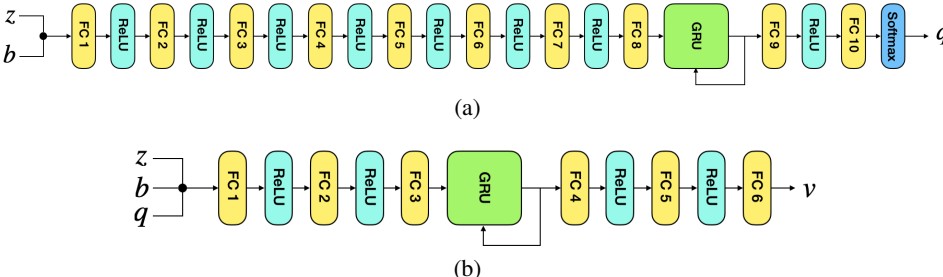

(a)

(b)

Figure 5: Network architectures of (a) the attention generator and (b) the critic function.

- observable states of 20 nearest exo-agents, $s_{exo} = \{p_i, v_i, \alpha_i\}_{i=1,\dots,20}$, where $p_i$, $v_i$, $\alpha_i$ are the position, velocity, and heading direction of the $i$th exo-agent.
- hidden states of 20 nearest exo-agents, $\theta_{exo} = \{\theta_i\}_{i=1,\dots,20}$, where $\theta_i$ is the intention of the $ith$ exo-agent. Suppose an exo-agent has $M$ potential paths to undertake according to the lane network, the value of its intention $\theta$ will be taken from $\{0, \dots, M-1\}$.

A belief $b$ is thus a discrete probability distribution defined over the hidden states or intentions of exo-agents, assuming probabilistic independence between different participants. It is represented using $\sum_{i=1}^{20} M_i$ probability values, where $M_i$ is the number of intentions for the $i$th exo-agent. An importance distribution $q$ is specified in the same way.

- **Action Modeling**: An action $a$ of the ego-vehicle is its acceleration discretized to three values, *ACC*, *CUR*, and *DEC*, meaning to accelerate, keep the current speed, and decelerate. The acceleration and deceleration are $3m/s^2$ and $-3m/s^2$, respectively.

- **Observation Modeling**: An observation $z$ from the environment includes all observable parts of the state $s$ and excludes the hidden intentions. Namely, $z = (s_c, s_{exo})$. Due to perceptual uncertainty, these observations often come with noise. However, in this work, we particularly focused on the uncertainty in human behaviors and ignored perceptual uncertainty, because the latter often has a secondary influence on decision-making.

- **Transition Modeling**: Our transition model assumes the ego-vehicle follows its reference path using a pure-pursuit steering controller and the input acceleration. Exo-agents are not controlled by the algorithm. We assume they take one of their hypothetical intended paths, using the GAMMA motion model [31] to interact with surrounding participants. When simulating an exo-agent, the GAMMA model conditions prediction on a set of factors, including the intended path, collision avoidance with neighbors, and kinematic constraints. At each time step, all agents are simulated forward by a fixed duration of $1/3s$. Afterward, small Gaussian noises are added to all transitions to model uncertain human control. So, the transition function of the ego-vehicle and exo-agents can be written as:

$$s'_c = \text{BicycleModel}\left(s_c | acc = a, steer = \text{PurePursuit}(s_c, P_c)\right), \tag{14}$$

where $s_c$ is the state of the ego-vehicle, $P_c$ is its planed reference path, and

$$s'_{exo} = \text{GAMMA}(s_c, s_{exo}) + \epsilon, \tag{15}$$

where $s_{exo}$ represents the states of ego-vehicles, and $\epsilon$ is a sampled Gaussian noise.

- **Reward Modeling**: The reward function takes into account driving safety, efficiency, and smoothness. When the ego-vehicle collides with any exo-agent, it imposes a severe penalty of $r_{co} = -20 \times (v^2 + 0.5)$ depending on the driving speed $v$. To encourage driving smoother, we also add a small penalty of $r_{sm} = -0.1$ for the actions *ACC* and *DEC* to penalize excessive speed changes. Finally, to encourage the vehicle to drive at a speed closer to its maximum speed $v_{max}$, we give it a penalty of $r_{ef} = \frac{v - v_{max}}{v_{max}}$ at every time step. The above rewards are additive. So,

$$r(s, a) = r_{co}(s) + r_{ef}(s) + r_{sm}(a) \tag{16}$$

