# OpenReview forum: "LEADER: Learning Attention over Driving Behaviors for Planning under Uncertainty"
_robot-learning.org/CoRL/2022/Conference — CoRL 2022 Oral_

### Official Review · Reviewer_D38s · 2022-07-12

**Originality:** Good
**Technical Quality:** Good
**Clarity Of Presentation:** Good
**Impact:** 2

**Recommendation:**

Weak Accept: I recommend accepting the paper, but will not argue for my recommendation if the majority of other reviewers have a different opinion.

**Summary:**

The paper indroduces an algorithm that uses an adversarial set up to learn autonomous driving behaviours from Monte Carlo-based partially observable reinforcement learning. It uses importance weighting to deal with the issue of rare, but safety-critical, events being missed in sampling. The performance of the approach is evaluated in three simulated driving environments and compared against a number of competing algorithims.

**Issues:**

- The curves in Figure 3 only show performance in one (carefully selected? representative?) test run. Please overlay these on a plot showing the average curve with error bars.
- I couldn't understand what the video was meant to show - perhaps because it is missing its sound? Please add a commentary and/or subtitles to explain what is going on with it.
- Please clarify how the test runs were apportioned between each map in Table 1.
- Please specify what 'average' is reported in Table 1. (Is it mean performance?) Moreover, please specify how the errors are calculated. (Is it standard deviation?)


**Quality Of The Limitations Section:**

Limitations are addressed clearly

**Reviewer Expertise:**

2: The reviewer is willing to defend the evaluation, but it is quite likely that the reviewer did not understand central parts of the paper

**Robotics Focus:**

Relevant but unlikely to deploy to hardware in near future

**Strengths And Weaknesses:**

Strengths:
- It deals with an important and challenging problem.
- The adverserial setup appears to lead to robustness.
- The mathematical development seems appropriate to the problem the authors are trying to address.
- It presents results that are competitieve to the state of the art in simulation.
- The discussion of limitations is good.

Weaknesses:
- The advance seems somewhat incremental. The results for LEADER in Table 1 are within a standard deviation of DESPOT-Uniform, so the improvement is probably not statistically significant.
- The evaluations are somewhat superficial in that they do not characterise the impact of the design decisions made in developing the proposed approach, or its sensitivity to the parameters used.
- It is doubtful that the results reported will scale to real world situations with a physical robot.


**Summary Of Recommendation:**

Overall, the paper is of good quality, and addresses an important problem. It presents a reasonable approach that seems appropriate to the problem in hand. I did not detect any issues with the mathematical development. The results presented show that the approach reaches performance comparable with the state of the art, and possibly slightly better (although the statistical significance of this is questionable). The scalability to real world driving is dubious.

---

> ### Author Response · Authors · 2022-08-21
> **Weakness: Probablitcis significance of improvements over DESPOT-Uniform in Table 1**
>
> We apologize for the probabilistic significance issue. We have now conducted 400 runs for all algorithms in Table 1, so that the results become probabilistically significant. The updated numbers show that the performance of LEADER is significantly better than DESPOT-Uniform, in terms of its collision rate, driving efficiency, and smoothness.

---

> ### Author Response · Authors · 2022-08-21
> **Weakness: Analysis on the design decisions and sensitivity to parameters**
>
> We agree with the importance of further analyses. Due to the space constraint, we were not able to present a thorough analysis of design choices, such as the choice of NN architectures, use of the warm-up training phase, and the sensitivity to parameters, such as the number of samples used in tree search, parameters of the reward function, etc. These analyses require retraining of the entire algorithm, which may not be feasible during the rebuttal phase. Nevertheless, we will certainly provide these analyses in our journal version.

---

> ### Author Response · Authors · 2022-08-21
> **Weakness: Scale to real-world situations with a physical robots**
>
> We recognize that it requires substantially more careful modeling efforts, more sophisticated system designs, and probably also real-world data, for such an algorithm to work on physical robots, which is unfortunately beyond the capability of typical university research teams. However, the planning and learning algorithm presented here is general, not constrained to a specific POMDP model, training and testing environment, or a specific system design. Therefore, we believe it will benefit the development of autonomous vehicles from an algorithmic perspective. Related approaches have been successfully deployed on real robot vehicles:
>
> E. Galceran, A.G. Cunningham, R.M. Eustice, and E. Olson, 2017. Multipolicy decision-making for autonomous driving via changepoint-based behavior prediction: Theory and experiment. Autonomous Robots, 41(6), pp.1367-1382.

---

> ### Author Response · Authors · 2022-08-21
> **Issue: Number of test runs and missing error bars in Figure 3**
>
> Figure 3 shows the average performance of 10 test runs. We periodically evaluate the algorithm using 10 randomly initialized runs during training. We have now added the error bars to Figure 3.

---

> ### Author Response · Authors · 2022-08-21
> **Issue: Video subtitles**
>
> We have now added subtitles and annotations to the video to explain better what goes on.

---

> ### Author Response · Authors · 2022-08-21
> **Issue: Table 1: how the test runs were apportioned between each map, the meaning of “average” and “errors”**
>
> The test runs for Table 1 were equally apportioned between the three maps. At each run, we randomly choose a map and a map setup using equal probabilities. “Average” is the mean performance. Errors are standard deviations. We have now added these explanations to the manuscript. Please see the attachment of the overall response.

---

> ### Comment · Reviewer_D38s · 2022-08-25
> **Overall reviewer response to revisions**
>
> This reviewer would like to thank the authors for their responses and revisions to the manuscript.
>
> The changes made in the evaluations section substantially improve the clarity and increase my faith in the significance of the results. The video is now also much more informative.
>
> I already scored this paper as a weak accept and will keep it unchanged.

---

### Official Review · Reviewer_KBsD · 2022-07-21

**Originality:** Fair
**Technical Quality:** Fair
**Clarity Of Presentation:** Good
**Impact:** 2

**Recommendation:**

Weak Accept: I recommend accepting the paper, but will not argue for my recommendation if the majority of other reviewers have a different opinion.

**Summary:**

This paper introduces a method for planning paths around other agents whose intentions are not directly observable. The contribution is in the use of an attention network to extract some information about the other agents, which is fed into a POMDP planner to select the ego vehicle's actions. Experiments on three simulated traffic scenarios suggest the proposed method outperforms methods based on Monte Carlo POMDP planners, model-based RL and model-free RL in terms of collision rate, distance traveled, etc.

**Issues:**

A new problem formulation would likely require a completely new algorithm, which seems beyond the scope of the rebuttal phase.

**Quality Of The Limitations Section:**

Limitations are addressed clearly

**Reviewer Expertise:**

4: The reviewer is confident but not absolutely certain that the evaluation is correct

**Robotics Focus:**

Relevant but unlikely to deploy to hardware in near future

**Strengths And Weaknesses:**

Strengths:
- The problem setting is very important, challenging, and relevant to the CoRL community
- The method seems to outperform the chosen baselines on the simulated driving tasks

Weaknesses:
- It's not clear that the problem has been structured in a way that can directly impact modern frameworks for this problem. Modern systems use advanced trajectory prediction algorithms on other agents (this paper has a relatively simple 3 discrete intentions), plan a detailed vehicle trajectory to be tracked according to the vehicle dynamics (the ego vehicle's 3 actions in this paper are accelerate, decelerate, no-op), and involve collision checking (is it embedded in the reward function? I couldn't find it in the paper), for example.
- The POMDP formulation is written out too generally, with the important details relevant to this problem left out: what are the models used for O, T, R for this specific problem and how do we know these are good models for multiagent collision avoidance?

**Summary Of Recommendation:**

The problem formulation is too simplified to demonstrate that the proposed technique is relevant to the real issues here.

---

> ### Author Response · Authors · 2022-08-21
> **Weakness: Whether the problem structure can impact modern frameworks for this problem: plan detailed trajectories, advanced trajectory prediction, number of intentions, collision checking**
>
> There is likely a misunderstanding on our technical claim. We claim an algorithm that learns behavioral attentions for planning under uncertainty. We do not intend to claim contribution for the particular POMDP model nor the overall AD system we used for experiments. Our POMDP planning framework can incorporate more sophisticated trajectory prediction models, dynamics models, and control spaces. Our planning and learning algorithms are general and independent of the specific system architecture.
>
> Further, our experimental system does cover the aspects of modem frameworks that the reviewer mentioned.
>
> - The system does plan for detailed trajectories for the AV in a decoupled manner. A path planner proposes a reference path; the POMDP planner determines acceleration along the reference path. The path planner is not part of our claimed algorithmic contribution, therefore, not discussed in detail. The POMDP planner provides closed-loop plans for acceleration, where future accelerations are conditioned on future observations, which are more reliable than open-loop plans.
>
> - Moreover, our POMDP model does use an advanced trajectory prediction algorithm for exo-agents, which is a traffic agent motion model [22] that considers both the intention of an exo-agent and collision avoidance interactions with its neighbors. The motion model also considers nonholonomic vehicle dynamics. The prediction trajectories used by the search tree are interactive, and more complex than the intention paths visualized in Figure 4 and in the video. The interactive trajectories are difficult to visualize, as they depend on future robot actions. Therefore, in Figure 4 and in the video, we visualized the simpler intention paths for illustration purposes only.
>
> - The intention sets of exo-agents are not limited to 3 intentions. They are extracted from the lane network, capturing representative behaviors allowed by the connectivity of the road.
>
> - Our algorithm also involves collision checking. Collision detection is done for every simulated state in the search tree. Collision checking results are reflected in the reward (Appendix C).
>
> We have improved our manuscript to clarify the above aspects. Please see the attachment of the overall response.

---

> > ### Comment · Reviewer_KBsD · 2022-08-21
> > **Thank you for the explanations**
> >
> > Thank you for clarifying -- the additions in Appendix C address my concerns about the modeling details and how the method is incorporated into to a full system.
> >
> > It would be nice if the video could be extended, solely to show maybe a few minutes of uninterrupted driving by the agent in some scenes. The annotations and pausing are helpful to illustrate individual issues, but there are some peculiar moments that make me wonder how smooth the behavior is over a longer timescale. For example, ~1:45 after the two cars in front have left, the agent stays still for a few seconds clogging the entrance. Around 2:40 the ego agent oscillates a few times between speeding up and slowing down (fairly aggressively) relative to the gray car in front of it.
> >
> > I will raise my score to "Weak Accept"

---

> > > ### Author Response · Authors · 2022-08-23
> > > **Thanks for your response**
> > >
> > > Thank you for raising the score.
> > >
> > > Please see the uninterrupted version of the previously shown clips and a few additional driving videos at this link: www.dropbox.com/s/mg8z7p0ff88pyo5/compiled_driving_videos_s.mp4?dl=0
> > >
> > > Delay of moving off and oscillation of speed happens sometimes, because we only used a fixed acceleration for both speeding up and slowing down and executed them using a simple PID controller. To achieve smoother speed control, the POMDP model can use high-level options such as “follow the front car”, “drive at x km/h”, “emergency brake”, etc, as actions, together with more accurate speed control methods to execute them. The same algorithm directly supports different action space designs.

---

> > > > ### Comment · Reviewer_KBsD · 2022-08-23
> > > > **Got it, thanks!**
> > > >
> > > > -

---

> > > > > ### Author Response · Authors · 2022-08-27
> > > > > **Update of score**
> > > > >
> > > > > Dear reviewer, since the rebuttal phase is soon ending, can you help to update your score as mentioned?

---

> ### Author Response · Authors · 2022-08-21
> **Weakness: The POMDP formulation is written out too generally**
>
> We appreciate your feedback. We have now improved the presentation of Appendix C to state the model elements more concretely, in the form of mathematical equations and with more details. Please see the revised Appendix C.

---

### Official Review · Reviewer_4p6j · 2022-07-28

**Originality:** Very Good
**Technical Quality:** Excellent
**Clarity Of Presentation:** Excellent
**Impact:** 4

**Recommendation:**

Strong Accept: I recommend accepting the paper and will argue for my recommendation even if other reviewers hold a different opinion.

**Summary:**

This paper proposes an importance sampling based planning algorithm in which the importance sampling is learned using neural networks to achieve safe and efficient driving. As opposed to existing work where importance sampling is not time-efficient, the proposed algorithm LEADER learns the distribution for importance sampling. This distribution is trained such that it is biased towards generating samples that are adversarial (against the ego vehicle). A Monte-Carlo based planner then uses these adversarial samples to come up with conservative best driving actions. Simulation experiments show the proposed algorithm outperforms the baselines in terms of safety.

**Issues:**

- An additional analysis I suggest authors to consider (possibly as a future work) is about how to use importance sampling. So the attention generator network is able to output the distribution for importance sampling. If my understanding is correct, when the planner tree is constructed, it is only the initial state that is being sampled from this attention generator network. The rest of tree is again populated using the distribution $p(s_{t+1}, z_{t+1} \mid s_t, a_{t+1})$, which is also implied by Equation 4. Now, one could also perform importance sampling for the time steps that are further in the future: for example, instead of just the initial state, one could sample the subsequent two time steps from this distribution. On the extreme, the tree would be constructed such that all transitions come from the importance sampling distribution. I guess this would cause the ego agent to try to plan as if all vehicles will attempt to crash with it, so it would be super conservative. By tuning how many time steps are sampled from the importance sampling distribution, the conservatism of the planner could be tuned. I wonder if this tuning could result in even better performance and/or higher safety.
- I am a little confused about the training procedure for the attention generator. What prevents it from outputting a distribution whose samples are dynamically infeasible transitions, e.g., an exo-vehicle teleporting just in front of the ego vehicle? What is the component (of the network or the loss function) that makes sure the samples from $q$ are dynamically feasible?
- The title of the paper is a little misleading, because there are really works that try to learn the drivers' attention to better predict their trajectories. See, for example, "Social LSTM: Human Trajectory Prediction in Crowded Spaces" by Alahi et al. (even though it is for pedestrians), or "Leveraging Smooth Attention Prior for Multi-Agent Trajectory Prediction" by Cao et al. (which is for driving).
- I found two typos: (i) In the caption of Figure 1, "The learning components includes" has a singular-plural mismatch, (ii) in Section 7, "This work make a few assumptions" has the same issue.

**Quality Of The Limitations Section:**

Limitations are addressed clearly

**Reviewer Expertise:**

4: The reviewer is confident but not absolutely certain that the evaluation is correct

**Robotics Focus:**

Highly relevant to robotics but no hardware experiments

**Strengths And Weaknesses:**

Strengths:
- The paper is very well written.
- The proposed approach makes a lot of sense and is not completely based on "neural network magic" -- it is very intuitive that importance sampling that biases search towards risky scenarios would improve safety in autonomous vehicles and this paper is using deep learning techniques only to accelerate the importance sampling procedure.
- Experiments (despite being simulated) show significant improvements in realistic scenarios.

Weaknesses:
- There is not really a major weakness, but the paper could get even better with some additional analysis on how to use importance sampling. See my detailed comments.

**Summary Of Recommendation:**

This paper does a great job in various terms: (i) it is proposing a very intuitive, easy to understand algorithm, (ii) it accelerates the algorithm using machine learning techniques and the way it achieves this (the training procedure) is not trivial, (iii) it has excellent results even though the experiments are all in simulation, which is quite common in autonomous driving research due to high costs and safety issues.

---

> ### Author Response · Authors · 2022-08-21
> **Issue: Importance sampling for further time steps**
>
> This is a very interesting point! Indeed, learning to sample subsequent transitions, not only the initial states, is an important direction to extend this work. The reviewer’s idea of learning for a few more steps is an interesting possibility. One challenge here is scalability, as the number of distributions to output would grow exponentially with the number of steps. One of our planned future directions is to design an algorithm that learns subsequent sampling distributions efficiently.

---

> ### Author Response · Authors · 2022-08-21
> **Issue: What prevents the generator from outputting a distribution whose samples are dynamically infeasible transitions**
>
> This is a great point for us to clarify more. We used a traffic agent motion model [22] to simulate how exo-agents actually execute a sampled intention (a path retrieved from the lane network). The motion model constrains predictions on a set of factors, including the agent’s intended path, collision avoidance with neighbors, and kinematic/dynamics constraints. Therefore, all predicted transitions are dynamically feasible. We have now improved the presentation of the transition function in Appendix C to avoid similar confusion. Please see the attachment of the overall response.

---

> ### Author Response · Authors · 2022-08-21
> **Issue: The title of the paper is a little misleading because there are works that learn attentions for predict trajectories**
>
> Thanks for raising this concern. To differentiate from the mentioned works, we now changed our title to “LEADER: Learning Attention Over Driving Behaviors For Planning Under Uncertainty”.

---

> ### Author Response · Authors · 2022-08-21
> **Fix of typos**
>
> Thanks for pointing out the typos. We have fixed them.

---

### Official Review · Reviewer_Djif · 2022-07-31

**Originality:** Good
**Technical Quality:** Good
**Clarity Of Presentation:** Very Good
**Impact:** 3

**Recommendation:**

Weak Accept: I recommend accepting the paper, but will not argue for my recommendation if the majority of other reviewers have a different opinion.

**Summary:**

This paper addresses the problem of learning amidst uncertainty in human behaviors.  Monte Carlo (MC) sampling techniques leveraged in addressing the attendant POMDP problem are often used for online planning due to their “anytime” properties.  To reason on risky / critical scenarios, techniques such as IS-DESPOT scheme (introduced by Luo, et. al. [9]), use importance sampling to bias MC sampling toward these risky scenarios, however, to achieve safe performance (e.g. low autonomous vehicle collision rates), the importance sampling distribution can be learned.  The technical contribution of this paper is to learn these importance distributions via a min-max game played between the importance distribution generator and the planner.  Comparisons are drawn between the proposed approach and existing approaches (DESPOT, other RL approaches) in various crowded driving scenarios.

**Issues:**

There are two main issues to be addressed here.

1. The authors are strongly encouraged to consider related approaches that were missed in their related work (outlined above), and discuss how their approach contrasts with these.

2. The assumption of pre-defined discrete intents is limiting in practice.  Is there a principled way of generating these intents?  What is the loss in generality by adopting a particular choice of discrete intents?  What are the limitations when applied to the continuum of actions and intents?

**Quality Of The Limitations Section:**

Limitations are addressed clearly

**Reviewer Expertise:**

4: The reviewer is confident but not absolutely certain that the evaluation is correct

**Robotics Focus:**

Highly relevant to robotics but no hardware experiments

**Strengths And Weaknesses:**

The paper offers a solution to an important problem in planning, which amounts to a combination of importance sampling-based Monte-Carlo planning approach with a learned importance distribution (attention).  While the approach to learning importance distributions is not new (acknowledged by the authors and further highlighted in the references below), the paper is well-written, and offers some ideas on framing the problem, including setting up an actor-critic-like scheme for training attention and critic.  It also provides good insight into the construction of the critic and attention networks (via the supplement).

All the descriptions are fairly complete and the supplement provides sufficient detail for reproducibility.  The limitations section is also very thoughtful and complete.

The problem is this well-explored field; to my estimation, the biggest limitation of the paper is that the related work is limited and lack comparison to relevant approaches that solve essentially the same problem (i.e. using a variation of importance sampling).  Some relevant works are below:

[A] Matthew O'Kelly, Aman Sinha, Hongseok Namkoong, Russ Tedrake, John C. Duchi: Scalable End-to-End Autonomous Vehicle Testing via Rare-event Simulation. NeurIPS 2018: 9849-9860

[B] Mansur Arief, Zhiyuan Huang, Guru Koushik Senthil Kumar, Yuanlu Bai, Shengyi He, Wenhao Ding, Henry Lam, Ding Zhao: Deep Probabilistic Accelerated Evaluation: A Robust Certifiable Rare-Event Simulation Methodology for Black-Box Safety-Critical Systems. AISTATS 2021: 595-603

[C] Edward Schmerling, Marco Pavone: Evaluating Trajectory Collision Probability through Adaptive Importance Sampling for Safe Motion Planning. Robotics: Science and Systems 2017

[D] Atrisha Sarkar, Krzysztof Czarnecki: A behavior driven approach for sampling rare event situations for autonomous vehicles. IROS 2019: 6407-6414

This gives rise to several basic questions: How do these approaches compare conceptually with the proposed one?  What is novel about the current approach?  Could any of these approaches be added to the empirical comparison?

Another fairly significant weakness relates to the assumption of a finite set of intentions and observations.  Such a representation is quite limiting in practice, as it seems to depend heavily on the chosen concept class (choice of discrete maneuvers / behaviors).  Presumably the simulated agents follow the same discrete behaviors, and hence the approach may not generalize well as a result.  Could the approach work with continuous intents?  What modifications would be necessary?

**Summary Of Recommendation:**

There is still some question I hope the authors can address in their revision surrounding the novelty of the importance sampling-based approach, as this has been extensively studied in the literature already.  Many such relevant works were missing in the paper.  Also, it is not clear whether the approach could apply to real-world driving as it only considers a finite set of discrete intents for agents.  Nonetheless, the idea of solving the problem via a critic-based approach and the techniques used are quite nice and would serve as a contribution to risk-awareness in planning.  I would therefore lean toward accept.

---

> ### Author Response · Authors · 2022-08-21
> **Issue: Missing references on the use of IS**
>
> Thanks a lot for the additional references. They are indeed related, at the high level, in terms of applying importance sampling to MC simulation for autonomous driving.
>
> However, these referenced works, in fact, have different technical objectives. They consider the problem of evaluating or verifying the safety of a given policy. This work focuses on the problem of planning for an optimal policy. The problem of policy verification has an offline nature. Given a policy, there is sufficient time to iteratively sample scenario sets, evaluate the effects on the tested policy, then iteratively update or learn the sampling distribution.
>
> Although planning can be considered as solving a collection of policy evaluation problems, the planning problem has an online nature. For a real-time situation, the algorithm needs to generate an importance distribution of human behaviors within milliseconds. Search-based methods [references A, C, D] or offline learning methods [reference B] for constructing importance distributions are thus not applicable here due to the constraint on the computation time.
>
> We have now added these additional references, as well as relevant discussions in the introduction section. Please see the attachment of the overall response.

---

> ### Author Response · Authors · 2022-08-21
> **Issue: The assumption of pre-defined discrete intents: principled ways to generate them, loss in generality, how to handle continuum of actions.**
>
> We think this is a valid concern and an interesting point to discuss. We consider abstraction over behaviors an important tool for improving computational efficiency. Successful applications of discrete abstractions of behaviors has been demonstrated on real self-driving cars:
>
> E. Galceran, A.G. Cunningham, R.M. Eustice, and E. Olson, 2017. Multipolicy decision-making for autonomous driving via changepoint-based behavior prediction: Theory and experiment. Autonomous Robots, 41(6), pp.1367-1382.
>
> To address the particular concerns on our algorithm:
>
> 1. Construction of the intention set.
> In the presented work, the intention set of an exo-agent is extracted by searching over the lane network. The set comprehensively covering sensible behaviors supported by the road structure, and changes according to the real-time location. A more general approach to constructing the intention set is to learn them from data. Data-driven models can learn multi-modal predictions, which correspond to multiple intentions. Although this technical direction is orthogonal to the planning problem investigated here, it is a promissing way to automatically generate intentions, including regular and irregular ones.
>
> 2. Continuous actions.
> We use a hierarchical and hybrid model to deal with continuous actions, combining discrete behavioral intentions at the high level and a continuous-action motion model for executing the intentions at the low level. The lower-level continuous-action model is part of the transition function presented in Appendix C.

---

### Official Review · Reviewer_Fx8v · 2022-08-04

**Originality:** Very Good
**Technical Quality:** Very Good
**Clarity Of Presentation:** Excellent
**Impact:** 4

**Recommendation:**

Strong Accept: I recommend accepting the paper and will argue for my recommendation even if other reviewers hold a different opinion.

**Summary:**

To manage uncertainty and risk-aware planning in the autonomous driving domain, the authors propose LEADR - a hybrid learning/planning method that solves the POMDP problem with learned behavior sampling distributions. To address the problem of missing critical events sampling in Monte Carlo sampling approaches, LEADER uses a neural network to generate sampling distribution that aims at minimizing the planner's expected return whereas the planner tries to maximize it's return conditioned on the learned sampling distribution. Similar to GAN, a minimax loss is designed for the neural network to learn to  sample critical/unsafe behaviors of other road agents. Evaluation is done in simulated (unregulated) urban driving environments and results suggest improved safety, efficiency and smoothness over comparison methods.

**Issues:**

- How does LEADER ensure that the network learns to sample behaviors that are critical but still reasonable (i.e. no abrupt changes in driving direction etc)

- If the sampled behaviors are all the more critical possibilities, does that make LEADER an overly conservative planner?

- Is there mode collapse during training?

**Quality Of The Limitations Section:**

Limitations are addressed clearly

**Reviewer Expertise:**

4: The reviewer is confident but not absolutely certain that the evaluation is correct

**Robotics Focus:**

Highly relevant to robotics but no hardware experiments

**Strengths And Weaknesses:**

The setup of the method reminds of generative adversarial imitation learning, but incorporation of a belief space planner instead of delegating everything to a neural network can improve sample efficiency (given good priors), safety and explainability. The paper is very well structured with key ideas clearly conveyed. The qualitative and quantitative results are convincing. The video is well-made and helpful in showcasing the capabilities of LEADER.

**Summary Of Recommendation:**

The proposed method is practical, theory is sound and evaluation results are convincing.

---

> ### Author Response · Authors · 2022-08-21
> **Issue: How does LEADER ensure sampled behaviors are reasonable?**
>
> This is a great point for us to clarify more. LEADER generates reasonable behaviors by leveraging the road context and a realistic traffic agent motion model. All candidate intentions are retrieved from the lane network of the map (using search), covering sensible behaviors supported by the road structure. Prediction of how an exo-agent will actually execute a parsed intention is generated using a traffic agent motion model [22], embedded in the POMDP transition function (Appendix C). We have improved our presentation in the overview and Appendix C to avoid similar confusion. Please see the attachment of the overall response.

---

> ### Author Response · Authors · 2022-08-21
> **Issue: Is LEADER an overly conservative planner**
>
> The reweighting mechanism of importance sampling (Eq. 5) prevents LEADER from becoming over-conservative. When estimating value, it considers the fact that critical events are deliberately sampled more frequently, so their importance weights are made smaller in order to still achieve an unbiased estimate of the reality.

---

> ### Author Response · Authors · 2022-08-21
> **Issue: Is there mode collapse during training**
>
> We used a simple trick to mitigate mode collapse, adding a smoothing factor to the distribution so that all intentions get a non-zero probability of being sampled. This is also related to the previous question---smoothing also prevents the planner from becoming over-conservative.

---

### Author Response · Authors · 2022-08-21
**Overall Response & Summary of Updates**

**Comment:**

We thank all reviewers for their thoughtful feedback. We are excited that reviewers found our work “practical, theory is sound and results are convincing” (R-Fx8v), “serves as a contribution to risk-awareness in planning” (R-Djif), “proposing a very intuitive, easy to understand algorithm” and the learning approach is “non-trivial” (R-4p6j), and solving an “important and challenging problem” (R-KBsD, R-D38s).

We have addressed all reviewer’s questions and concerns with rebuttal replies and paper revisions. The main updates in the revised paper are:

- We have added discussion on the additional references outlined by reviewer Djif, on their connection with and essential differences from our work.

- We have now clarified relevant details of our POMDP model and the AD system to address reviewer KBsD’s concern about the reality/validity of the problem setting, and the question on the dynamical feasibility of predictions raised by reviewer Fx8v and reviewer 4p6j.

- We have improved our quantitative and qualitative results according to the comments of reviewer D38s, including error bars in Figure 3, the probabilistic significance of numbers in Table 1, and video annotations.

Changes are highlighted in blue in the revised manuscript.


**Zip File:**

/attachment/4addce8c879a0520c4a4edea2bae6e7d74e9fb76.zip

---

### Meta-Review · Area_Chair_rwiG · 2022-08-15

**Recommendation:** Accept (Oral)
**Confidence:** 5

**Metareview:**

Strengths
- All the reviewers have found the proposed approach is technically sound.
- The evaluation has been done in a sufficiently fidelity and comprehensibility.
- The paper is clearly written.
- Literature has been strengthened during the rebutal phase.
- Concern on the problem setting has been well-clarified during the rebuttal phase.

**Best Paper Nomination:**

Yes